# Emerging Spiral Waves and Coexisting Attractors in Memductance-Based Tabu Learning Neurons

Balakrishnan Sriram [1], Zeric Njitacke Tabekoueng [2], Anitha Karthikeyan [3] and Karthikeyan Rajagopal [1,*]

[1]  Centre for Nonlinear Systems, Chennai Institute of Technology, Chennai 600069, India
[2]  Department of Electrical and Electronic Engineering, College of Technology (COT), University of Buea, Buea P.O. Box 63, Cameroon
[3]  Department of Electronics and Communications Engineering, University Centre for Research and Development, Chandigarh University, Mohali 140413, India
*  Correspondence: rkarthiekeyan@gmail.com

**Abstract:** Understanding neuron function may aid in determining the complex collective behavior of brain systems. To delineate the collective behavior of the neural network, we consider modified tabu learning neurons (MTLN) with magnetic flux. Primarily, we explore the rest points and stability of the isolated MTLN, as well as its dynamical characteristics using maximal Lyapunov exponents. Surprisingly, we discover that for a given set of parameter values with distinct initial conditions, the periodic and the chaotic attractors may coexist. In addition, experimental analysis is carried out using a microcontroller-based implementation technique to support the observed complex behavior of the MTLN. We demonstrate that the observed numerical results are in good agreement with the experimental verification. Eventually, the collective behaviors of the considered MTLN are investigated by extending them to the network of the lattice array. We discover that when the magnetic flux coupling coefficient is varied in the presence of an external stimulus, the transition from spiral waves to traveling plane waves occurs. Finally, we manifest the formation of spiral waves in the absence of an external stimulus in contrast to previous observations.

**Keywords:** tabu learning neurons; bifurcation; Lyapunov exponents; spiral waves

## 1. Introduction

The brain, which is made up of billions of neurons, is one of the most intricate systems in living creatures [1]. A fraction of neurons must communicate with one another to transmit the signals to the brain to perform various cognitive activities such as controlling consciousness, memory, emotion, touch, etc. [2]. The study of computational neuroscience can aid in the understanding of how brain systems work to execute such cognitive functions [3,4]. Therefore, the researchers are engaged in studying brain functions in diverse contexts. As a consequence, several biological neuron models, such as Hodgkin–Huxley (HH), FitzHugh–Nagumo (FHN), Hindmarsh–Rose (HR), Morris–Lecar (ML), and Izhikevich (IZH), to name a few, have been implemented and investigated to mimic the functioning of neurons in the brain [5,6]. It is revealed from the earlier reports that such neurons can exhibit various kinds of spiking, firing, and bursting patterns [7,8]. Further, the emergence of various collective behaviors has been reported, while the interacting collection of such neurons includes synchronization, desynchronization, chimera, clustering, and traveling wave patterns [9–13].

In addition to the neuron model described above, another kind of neuron model is the tabu learning neuron (TLN), which was proposed by Beyer and Ogier in 1991 [14]. This model is distinctive in that it conducts tabu searches in a solution space. Originally, the dynamical behavior of the single and two tabu neurons was reported, and this identified the occurrence of Hopf bifurcation [15,16] as varying the critical parameter of the system.

Following that, the occurrence of Pitchfork bifurcation, Flip bifurcation, and Neimark–Sacker bifurcation were demonstrated using the discrete TLN [17]. Furthermore, by using the sum of two delays as a control parameter in TLN, nonlinear behavior and the presence of Hopf bifurcation have been documented [18]. Additionally, many circuit implementations and experimental confirmations have been carried out to comprehend the dynamical characteristics of the TLN [19]. For instance, the multisim circuit simulation was carried out to mimic the nonlinear behavior of TLN using nonlinear approximation. Additionally, the existence of various complex activities including chaotic, periodic, spiking, bursting, and firing patterns and the coexisting attractors with bistability have been demonstrated using FPGA-based neuron circuits [20]. To explore the multistability properties with potential applications to medical picture encryption, an analog circuit for the tabu learning two-neuron model with compound activation has recently been constructed under the PSPICE environment [21].

Spiral waves (SW) are promising phenomena that can arise in a variety of realistic situations such as biological excitable media, amoebae colonies, fungus, growing crystals, chemical processes, fluids and gas eddies, and so on [22–25]. A self-sustaining rotating spiral wave, for instance, might arise abruptly in cardiac or brain tissue, damaging their dynamics and resulting in life-threatening illnesses. There have been reports of the spontaneous formation of spiral and plane waves in a mammalian neocortex [26–28], a turtle's visual cortex [29,30], and a chicken's retina [31], among other places. Recognizing the possible situations or conditions for the commencement of spiral waves is thus of tremendous theoretical significance, with several practical applications. The formation of spiral wave patterns was first seen in the Belousov–Zhabotinskii reaction (BZ) [32]. Following that, several investigations were carried out to demonstrate the spiral wave patterns and comprehend their processes using various excitable media. For instance, J. G. Milton et al. have demonstrated the propagation of SW using the randomly connected integrate and fire neurons [33].

The initiation and development of a spiral wave were further observed by generating a block in the target wave using two-dimensional Hodgkin–Huxley neurons with nearest-neighbor interactions [34]. Using the multilayered Huber–Braun (HB) neuron model, the spatial pattern of the spiral wave was further examined by implementing the obstacles in parallel or perpendicular orientations [35]. Additionally, it was inferred how such dynamical behavior could be characterized in terms of spatioperiod and spatioenergy using various excitable media, such as modified Hindmarsh–Rose neuron models, hybrid neuron models, and Morris–Lecar neuron models [36]. Despite the fact that spiral wave patterns have been observed in many biological neurons, the occurrence of such a phenomenon in the modified tabu learning neuron (TCN) model remains unclear, and the impact of magnetic flux on TCN has not been explored. In this study, we consider tabu learning neurons with magnetic flux due to the presence of the flux effect in realistic situations. At first, we show the dynamical behavior and its transition of an isolated neuron using two-parameter analysis, bifurcation analysis using local maxima of the variables, and maximal Lyapunov exponents. Further, we also provide experimental evidence for the observed dynamical behaviors of the MTLN. Finally, we show the existence of spiral wave patterns while extending the lattice array of MTLNs. In particular, contrary to previous findings, the occurrence of spiral wave patterns is demarcated even in the absence of external stimuli.

The following is the remainder of the article: Section 2 introduces the considered model of the tabu learning neuron, and the subsection discusses the fixed points and their stability. The dynamical characteristics of the considered system are discussed in Section 3. In particular, we demonstrate the coexistence of periodic and chaotic attractors and their characterizations using the bifurcation analysis, Lyapunov exponents, and basin of attraction. In Section 4, the observed results are experimentally validated using a microcontroller implementation of the proposed tabu learning neuron. Finally, in Section 5, the emergence of spiral waves is inspected by extending it to the lattice network. Section 6 contains the concluding remarks.

## 2. Model Description

A very common model of a neuron network [19,37] for the *j*th neuron is mathematically defined by the relation

$$c_j \dot{v}_j = \sum_j \sigma_{jk} X_k + I_j - R_j^{-1} v_j, \tag{1}$$

where $v_j$, $c_j$, and $R_j$ are the *j*th neuron's state variables, capacitance, voltage, and resistance, respectively. The coupling strength between the *j*th and *k*th neuron is given by $\sigma_{jk}$ and $I_j$ is an input current to the *j*th neuron. Typically, the activation function plays an intriguing role in exhibiting complex behaviors in neurons [21]. Here the activation function is defined by the term $X_k$. In [2], the authors have modified the model (1) with a linear proximity function and this is given by the relation

$$\begin{aligned}
\dot{x} &= ax + bX + y + I_{ext}, \\
\dot{y} &= -cy - dX
\end{aligned} \tag{2}$$

where *x* and *y* are the state variables of the tabu learning neurons. The parameters a, b, c, and d denote the positive constant, synaptic weight, memory decay rate, and learning rate, respectively. The activation function *X* is expressed as

$$X = \varepsilon x e^{-(\varepsilon x)^2 / \sigma^2}; \varepsilon, \sigma > 0. \tag{3}$$

It should be mentioned that the activation function (3) is bounded and continuously differentiable in both positive and negative directions. To further improve this learning model, we have considered magnetic flux coupling and the new mathematical model can be defined as

$$\begin{aligned}
\dot{x} &= -ax + bX + y + I_{ext} - k_0 x M(\phi), \\
\dot{y} &= -cy - dX, \\
\dot{\phi} &= k_1 x - k_2 \phi,
\end{aligned} \tag{4}$$

where the flux coupling is defined by a memductance function $M(\phi) = \alpha + 3\beta\phi^2$ and the excitation current is considered as a sinusoidal time-varying signal defined as $I_{ext} = m \sin(2\pi f t)$, where m and f are the amplitude and frequency of the excitation current. $k_0$, $k_1$, and $k_2$ are the constants. The parameter values are fixed as a = 0.2, b = 0.3, m = 0.2, f = 1.0, $\varepsilon$ = 10.0, $\sigma$ = 0.4472, $k_1$ = 0.01, $k_2$ = 0.5, $k_0$ = 0.1, $\alpha$ = 0.1, $\beta$ = 0.01, c = 0.5, and d = 1.0 and the initial conditions are considered as [0;0;0].

*Rest Points and Their Stability*

Rest points play an important role in the study of artificial neural networks. They enable the highlighting of the dynamical behavior exhibited by a neuron or neural network as self-excited or hidden. Let us recall that self-excited attractors are generated from a nonlinear dynamical system with unstable equilibria [38,39]. In contrast, hidden attractors are generated from systems without equilibria or systems with state equilibria [40–42]. Furthermore, hidden attractors have an attraction basin that does not overlap with the vicinity of an equilibrium point.

The rest points of Equation (4) are obtained by solving the equation $\dot{x} = \dot{y} = \dot{\phi} = 0$. After some algebraic manipulations, the following is obtained.

$$\begin{aligned}
x_e &= \frac{k_2}{k_1} \phi_e \\
y_e &= \frac{-d}{c} \varepsilon x_e e^{-(\varepsilon x_e)^2 / \sigma^2} \\
m \sin(2\pi f t) &= a \frac{k_2}{k_1} \phi_e - b \left( \varepsilon \frac{k_2}{k_1} \phi_e \right) e^{-\left( \varepsilon \frac{k_2}{k_1} \phi_e \right)^2 / \sigma^2} - \frac{d}{c} \left( \varepsilon \frac{k_2}{k_1} \phi_e \right) e^{-\left( \varepsilon \frac{k_2}{k_1} \phi_e \right)^2 / \sigma^2} \\
&\quad + \frac{k_0 k_2}{k_1} \phi_e \left( \alpha + 3\beta\phi_e^2 \right)
\end{aligned} \tag{5}$$

Therefore, the rest points of the proposed tabu learning neuron are given as

$$P_e = \left[\frac{k_2}{k_1}\phi_e; -\frac{d}{c}\left(\varepsilon\frac{k_2}{k_1}\phi_e\right)e^{-\left(\varepsilon\frac{k_2}{k_1}\phi_e\right)^2/\sigma^2}; \phi_e\right]$$

(6)

$\phi_e$ in Equation (6) is obtained by solving the transcendental equation provided by the third line of Equation (5).

The graphical resolution of the third line of Equation (5) over time enables us to obtain the AC rest points of the investigated tabu learning neuron as shown in Figure 1. In that same figure, the DC rest points are obtained when t = 0. From that figure, for some discrete values of the time taken in the interval $t \in [-0.2, 0.2]$, the considered model has only the origin as a unique rest point.

$$J_M = \begin{bmatrix} -a + b\varepsilon e^{-(\varepsilon x_e)^2/\sigma^2} - \frac{2b\varepsilon^3 x_e^2 e^{-(\varepsilon x_e)^2/\sigma^2}}{\sigma^2} - k_0\left(3\beta\phi_e^2 + \alpha\right) & 1 & -6k_0\beta x_e\phi_e \\ -d\varepsilon e^{-(\varepsilon x_e)^2/\sigma^2} + \frac{2d\varepsilon^3 x_e^2 e^{-(\varepsilon x_e)^2/\sigma^2}}{\sigma^2} & -c & 0 \\ k_1 & 0 & -k_2 \end{bmatrix}$$

(7)

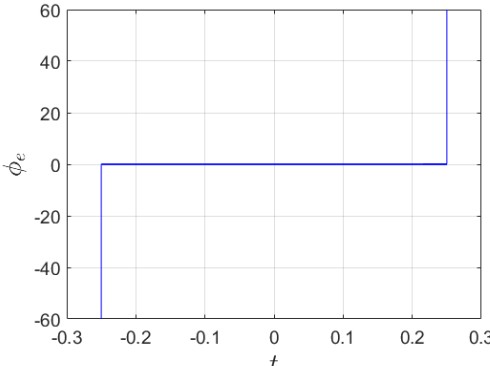

**Figure 1.** Time evolution of the AC equilibrium point of the considered tabu learning neuron model.

From the Jacobian matrix of Equation (7), the following third order characteristic equation is obtained

$$L(\lambda) = \lambda^3 + \eta_1\lambda^2 + \eta_2\lambda^1 + \eta_3 = 0$$

(8)

with the coefficients:

$$\eta_1 = k_2 + c - \left(-a + b\varepsilon e^{-(\varepsilon x_e)^2/\sigma^2} - \frac{2b\varepsilon^3 x_e^2 e^{-(\varepsilon x_e)^2/\sigma^2}}{\sigma^2} - k_0\left(3\beta\phi_e^2 + \alpha\right)\right)$$

$$\eta_2 = ck_2 - (c + k_2)\left(-a + b\varepsilon e^{-(\varepsilon x_e)^2/\sigma^2} - \frac{2b\varepsilon^3 x_e^2 e^{-(\varepsilon x_e)^2/\sigma^2}}{\sigma^2} - k_0\left(3\beta\phi_e^2 + \alpha\right)\right)$$

$$-k_1(-6k_0\beta x_e\phi_e) - \left(-d\varepsilon e^{-(\varepsilon x_e)^2/\sigma^2} + \frac{2d\varepsilon^3 x_e^2 e^{-(\varepsilon x_e)^2/\sigma^2}}{\sigma^2}\right)$$

$$\eta_3 = -k_1c(-6k_0\beta x_e\phi_e)$$

$$-k_2c\left(-a + b\varepsilon e^{-(\varepsilon x_e)^2/\sigma^2} - \frac{2b\varepsilon^3 x_e^2 e^{-(\varepsilon x_e)^2/\sigma^2}}{\sigma^2} - k_0\left(3\beta\phi_e^2 + \alpha\right)\right)$$

$$-k_2\left(-d\varepsilon e^{-(\varepsilon x_e)^2/\sigma^2} + \frac{2d\varepsilon^3 x_e^2 e^{-(\varepsilon x_e)^2/\sigma^2}}{\sigma^2}\right)$$

(9)

If the following inequalities are satisfied, the characteristic equation in Equation (8) indicates that the MTLN under consideration is stable:

$$\eta_1 > 0, \eta_2 - \frac{\eta_3}{\eta_1} > 0, \eta_3 > 0.$$

(10)

Therefore, for some discrete values of $k_0$, the eigenvalues of the MTLN as well as their stability are discussed in Table 1. From the eigenvalues analysis, it is obvious that the MTLN exhibits self-excited dynamics.

**Table 1.** Eigenvalues of the tabu learning neuron with their corresponding stability for some discrete values of the parameter $k_0$ around the origin.

| Values of $k_0$ | Eigenvalues | Associated Stability |
|:---:|:---:|:---:|
| 0.1 | $\lambda_1 = -0.5, \lambda_{2,3} = 1.145 \pm 2.7i$ | Unstable focus |
| 0.4 | $\lambda_1 = -0.5, \lambda_{2,3} = 1.13 \pm 2.709i$ | Unstable focus |
| 0.8 | $\lambda_1 = -0.5, \lambda_{2,3} = 1.11 \pm 2.722i$ | Unstable focus |
| 1.2 | $\lambda_1 = -0.5, \lambda_{2,3} = 1.09 \pm 2.733i$ | Unstable focus |

## 3. Dynamical Behaviors of Tabu Learning Neuron

The objective of this section is to investigate the complex behavior that can occur in the tabu learning neuron under consideration. The fourth-order Runge–Kutta algorithm is exploited to compute the nonlinear analysis tools used to characterize the model. For the accuracy of our calculations, variables and parameters of the MTLN are selected in extended precision mode. For each iteration, a fixed time step of $5 \times 10^{-3}$ is used.

### 3.1. Two-Parameter Analysis in Distinct Parametric Spaces

In order to explore the complex behaviors that can occur in the neuron model under consideration, the two-parameter diagrams have been computed in different parametric spaces. Those diagrams have been obtained by simultaneously varying two parameters of the proposed model and recording at each iteration the value of the maximum Lyapunov exponent of the model. From the value of the Lyapunov exponent of the considered neuron, two dominant behaviors have been recorded, as can be seen in Figure 2. On one hand, we have chaotic behavior supported by $\lambda_{max} > 0$. On the other hand, we have periodic behavior supported by $\lambda_{max} < 0$. For instance, in Figure 2a, the two-parameter diagram is depicted in $(k_0, f)$ parametric space, demonstrating that chaotic behavior can exist at sufficient magnitudes of f and $k_0$, while the rest of the parameters exhibit periodic behavior. Furthermore, in $(k_0, m)$ parametric space, chaotic behavior is observed only at lower magnitudes of excitation amplitude when $k_0$ is varied (see Figure 2b). Following that, Figure 2c,d depict the dynamical behavior of TLN in $(k_0, k_1)$ and $(k_0, k_2)$ parametric spaces, respectively. In both cases, chaotic attractors exist at lower $k_0$ values with all values of $k_1$. Increasing $k_0$ to larger values results in periodic behavior for entire parametric spaces. As a result, the obtained results outline the dynamical transitions in the various parametric spaces.

### 3.2. Coexisting Attractors through Bifurcation, Lyapunov Exponents, and Basin of Attraction

To characterize the local behavior of the proposed tabu learning neuron, the bifurcation diagrams of Figure 3a and their corresponding graphs of the maximum Lyapunov exponent of Figure 3b were used. In Figure 3, two main sets of data are superimposed, using the same set of parameters but starting from different initial conditions. For example, the set of data in red was obtained [0;0;0] by increasing the control parameter $k_0$, starting from the initial conditions. To obtain the set of data in red, the continuation technique was used. The final state of each iteration was used as the initial conditions for the next iteration.

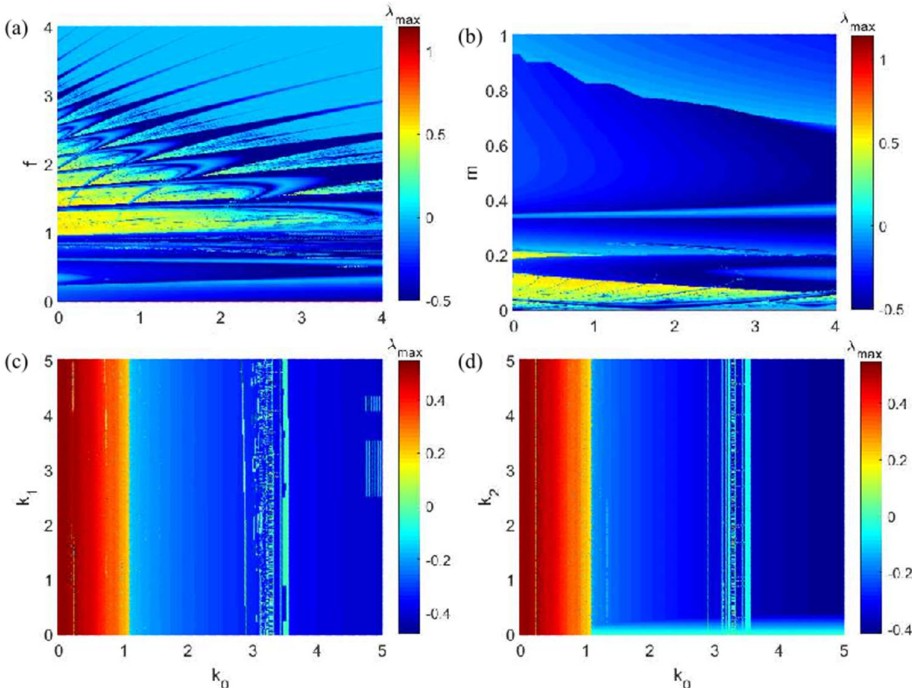

**Figure 2.** Two-parameter diagrams show the global dynamics of the proposed neuron model when two parameters simultaneously vary. (**a**) The plane ($k_0$, $f$) is obtained for $m = 0.2$, $k_1 = 0.01$, and $k_2 = 0.5$. (**b**) The plane ($k_0$, $m$) is obtained for $f = 1.0$, $k_1 = 0.01$, and $k_2 = 0.5$. (**c**) The plane ($k_0$, $k_1$) is obtained for $m = 0.2$, $f = 1.0$, and $k_2 = 0.5$, and (**d**) the plane ($k_0$, $k_2$) is obtained for $m = 0.2$, $k_1 = 0.01$, and $f = 1.0$. Initial conditions are [0;0;0].

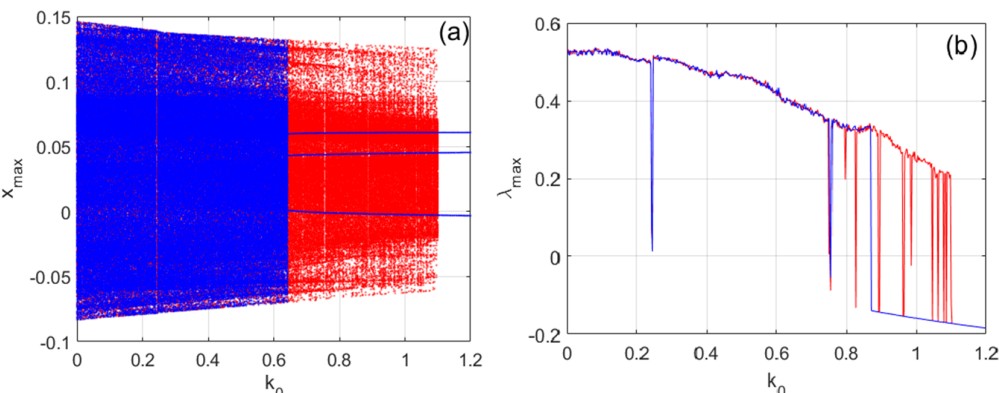

**Figure 3.** (**a**) Bifurcation diagram showing the local maxima of the state variable of the tabu learning neurons versus the electromagnetic induction strength $k_0$ and (**b**) shows the corresponding Lyapunov exponents as a function of $k_0$. The red color points/lines were obtained based on the continuation technique using initial conditions [0;0;0], while the blue ones were obtained with fixed initial conditions [0;0;6].

In contrast, the blue data set in Figure 3 was obtained by increasing the control parameter $k_0$ and starting each iteration from the same fixed initial conditions of [0;0;6]. It is good to emphasize that the superimposed bifurcation diagrams with their corresponding graph of the maximum Lyapunov exponent highlighted in this work are at the origin of the phenomenon of the coexisting attractors found in the proposed tabu learning neuron.

These coexisting attractors are further supported using the phase portraits of Figure 4a and their corresponding time series of Figure 4b for a discrete value of $k_0 = 1$. A basin of attraction can be defined as the set of initial conditions that give birth to a specific attractor. The basins of attraction of Figure 4c,d have been computed to characterize the domain of

the initial conditions related to each coexisting attractor of Figure 4a. From those attraction basins, it is obvious that the probability of obtaining the chaotic attractor (red domain) is too high compared to the probability of obtaining the periodic attractor (blue domain).

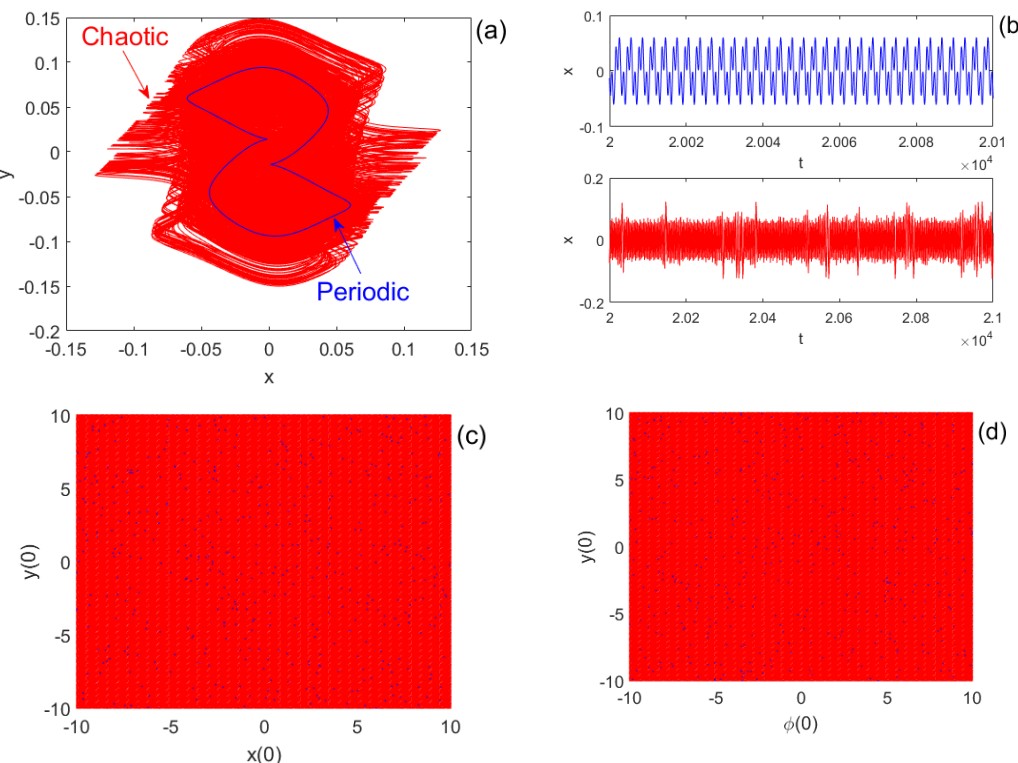

**Figure 4.** (**a**) Phase portraits of coexisting periodic and chaotic attractors and (**b**) their corresponding time series obtained with the initial conditions [0;0;0] for the chaotic behavior and [0;0;6] for the periodic behavior by fixing $k_0 = 1.0$. (**c**,**d**) represent the domain of the initial condition [x(0), y(0)] and [$\varphi$(0), y(0)] associated with each of the coexisting behaviors.

## 4. Microcontroller Implementation of the Proposed Tabu Learning Neuron

The microcontroller-based experiments offer more flexibility in order to bring the portable source code into practice, customize the system parameters, starting conditions, etc. It can also facilitate performing mathematical operations without the use of any special tools. Thus, we set up a system based on a microcontroller to experimentally mimic the observed coexisting dynamics of an MTLN. Figure 5a displays the corresponding schematic illustration. The microcontroller implementation is conducted in the first part using Arduino IDE and an ESP32-WROOM-32D board. For the sake of simplicity, speed, and accuracy, the ESP32 microcontroller from ESPRESSIF is used to discretize the system (3) and implement the tabu learning neuron model. Data acquisition can be conducted through the Arduino UNO. Finally, the output can be virtualized through the computer. The complete picture of the experimental setup for the microcontroller-based implementation of TLN is depicted in Figure 5b.

(a)

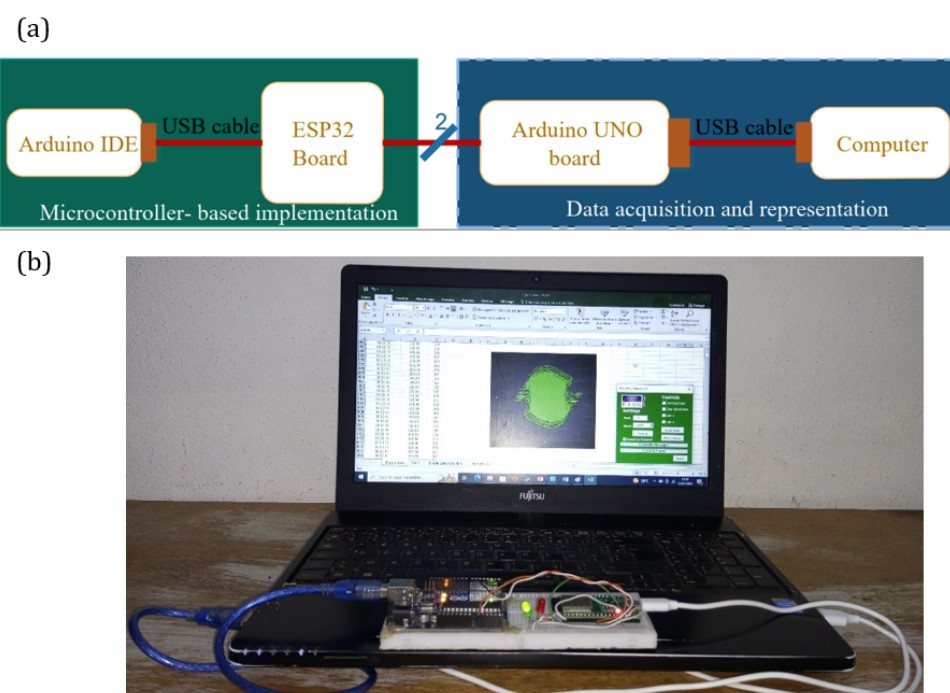

(b)

**Figure 5.** (**a**) Block diagram of the experimental setup and (**b**) the complete picture of the relevant experimental setup of microcontroller-based TLN model.

Figure 6 depicts the results of microcontroller implementation, which clearly demonstrate that the experimental setup of Figure 5 is capable of reproducing the coexisting behaviors observed in the tabu learning neuron model. By comparing Figure 6 with Figure 4a, the experimental coexisting phase planes obtained from the microcontroller implementation match well with the numerical simulation equivalents. In this way, the results of the microcontroller implementation help us to validate the dynamical characteristics of the proposed TLN model.

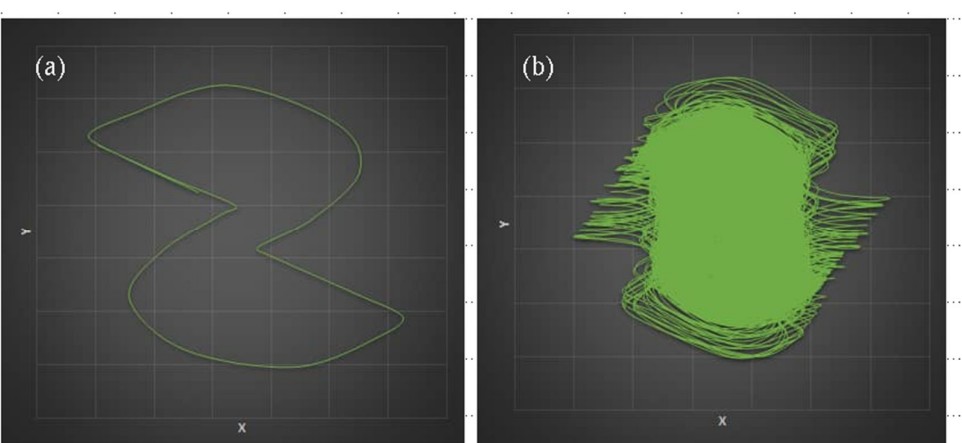

**Figure 6.** Experimental coexisting phase portraits obtained from the microcontroller implementation of the Tabu learning neuron. (**a**,**b**) are phase portraits of periodic and chaotic attractors, respectively, obtained with the same set of parameters given in Figure 4.

In the following sections, we extend the analysis to the network case as well based on our understanding of isolated Tabu learning neurons.

### 5. Network Dynamics of Tabu Learning Neurons

Many complex systems can exhibit a range of collective dynamical features in realistic situations when nodes interact with one another. As a result, exploring the dynamical behaviors of the neuron by considering the network with a large number of elements is intriguing. Thus, in order to outline the spiral wave and its transitions, we consider a lattice network of tabu learning neurons with closest neighbor connections. The associated dynamical equations are denoted as

$$\dot{x}_{jk} = -ax_{jk} + bX_{jk} + y_{jk} + I_{ext} - k_0 M\left(\phi_{jk}\right) x_{jk} + \sigma\left(x_{j+1k} + x_{j-1k} + x_{j\,k+1} + x_{j\,k-1} - 4x_{jk}\right) + g(t)\gamma_{jk},$$
$$\dot{y}_{jk} = -cy_{jk} + dX_{jk},$$
$$\dot{\phi}_{jk} = -k_1 x_{j\,k} - k_2 \phi_{jk}, \qquad j,\,k = 1,2,\ldots M. \tag{11}$$

where the total number of tabu neurons in the lattice array is defined by $M$ and the coupling strength is represented by the symbol $\sigma$. The external periodic stimulus described by the relation $g(t) = m\sin(2\pi ft)$ and applied at the center of the node in the lattice array. m and f are the amplitude and frequency of the external stimuli, which are fixed as m = 0.01 and f = 0.01, respectively. The periodic stimulus is distributed to the node when $\gamma_{j\,k} = 1$; otherwise, when $\gamma_{j\,k} = 0$, the periodic stimulus is not applied to the node. The dynamical behaviors of the system (11) are discussed in the following section for the presence and the absence of external stimuli in detail.

The dynamical behavior of the tabu network is initially explored in the presence of external stimuli by setting the coupling strength D = 1.0 for different values of $k_0$. When the magnitude of $k_0$ is low, multiple pools arise due to poor excitability nodes. As a result, the waves have a limited ability to propagate, resulting in spiral waves with many seeds, as illustrated in Figure 7 for $k_0$ = 0.001, $k_0$ = 0.01, $k_0$ = 0.05, $k_0$ = 0.1, and $k_0$ = 0.3. As $k_0$ increases, the nodes in the lattice become more pronounced, allowing for deeper wave penetration. As a consequence, we discovered that when $k_0$ = 0.5, $k_0$ = 0.8, and $k_0$ = 1.0, some of the weak pools disappear with the advent of a strong spiral with high excitability. When the magnitude of $k_0$ is increased, the neurons become more excitable, resulting in the elimination of all spiral seeds. Eventually, we observed the disappearance of spiral waves and the appearance of traveling plane waves for $k_0$ = 1.2.

According to earlier reports, external stimulation is required for the formation of spiral waves in excitable media [34–36]. Interestingly, we discovered that the considered tabu learning neurons display spiral waves even when no external stimulus is present, g(t) = 0.0, as shown in Figure 8, in contrast to the earlier reports. We observed low-intensity turbulent waves when $k_0$ = 0.01. Increasing the magnitude of $k_0$ enables spiral waves to emerge due to the deep propagation of waves with robust and low-intensity pools. As $k_0$ is raised, there is a noticeable decrease in heterogeneity, which results in the appearance of a traveling plane wave state for $k_0$ = 1.0. Ultimately, we discovered that spiral waves are entirely suppressed with the formation of traveling plane waves as $k_0$ is increased to larger values ($k_0$ = 1.2). Thus, from the finding, it is clear that the emergence of spiral waves and the transitions of the array of tabu learning neurons occurred in the presence and absence of the external stimulus. In particular, as compared to the previous findings, we observed the formation of SW even in the absence of external stimuli.

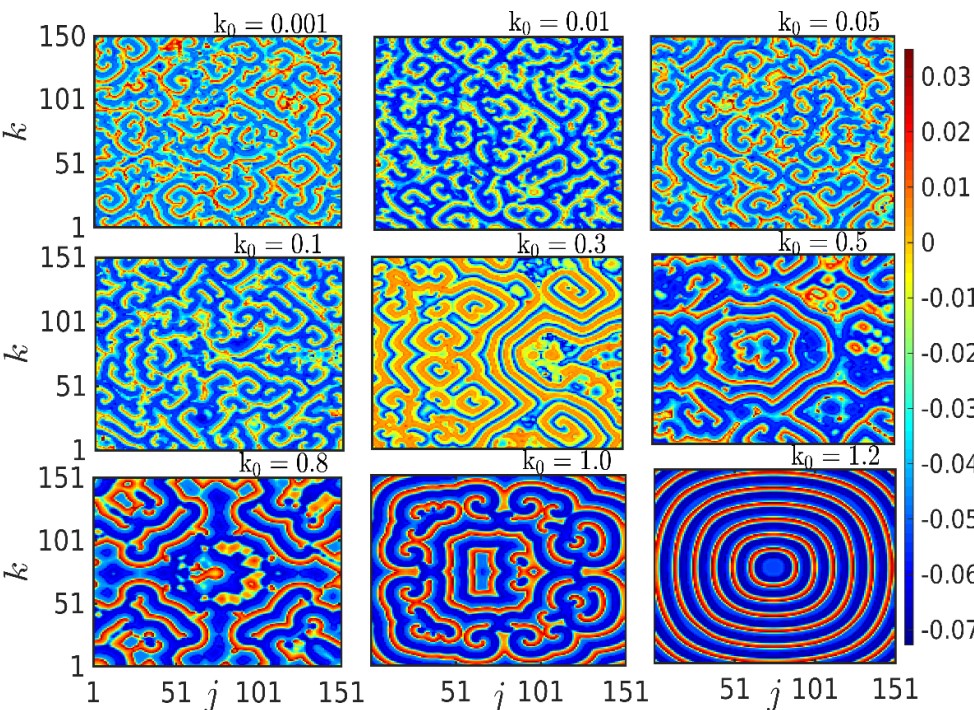

**Figure 7.** The transition from spiral waves to traveling plane waves in the presence of external stimuli when $k_0$ was increased to 0.001, 0.01, 0.05, 0.1, 0.3, 0.5, 0.8, 1.0, and 1.2. The other parameter values were fixed as specified in Section 2.

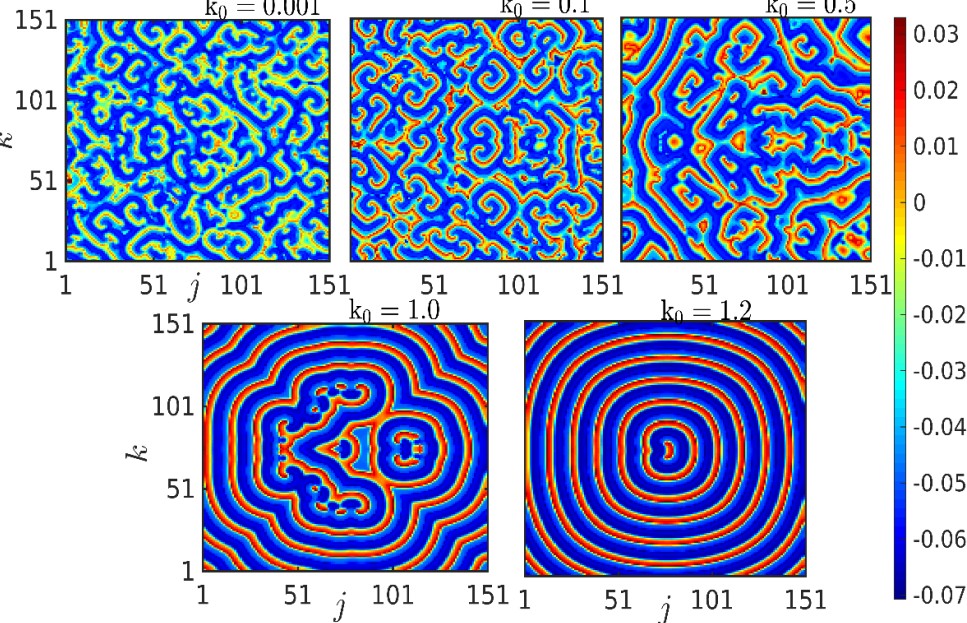

**Figure 8.** Spiral wave pattern formation and transition to traveling plane wave without external stimulus while increasing flux coupling coefficient as $k_0 = 0.001$, $k_0 = 0.1$, $k_0 = 0.5$, $k_0 = 1.0$, and $k_0 = 1.2$. The other parameter values were fixed as specified in Section 2.

## 6. Conclusions

In this study, we have considered a modified tabu learning neuron (MTLN) model with memductance-based flux coupling. We first addressed equilibrium points and stability of the MTLN. Following that, the two-parameter analysis was performed in different parametric spaces to discover the dynamical regimes. Subsequently, the transitions of

the attractors were inspected using bifurcation analysis, and we found the coexistence of periodic and chaotic attractors as a function of flux coupling strength. In the presence of magnetic flux, such coexisting attractors are not well explored in the MTLN model. Therefore, the coexisting attractors and their transitions were further substantiated using the Lyapunov exponents. Furthermore, the obtained results were experimentally validated using the microcontroller-based implementation technique. Finally, we extended our investigation to the lattice array of the MTLN network and discovered that raising the magnetic flux coupling coefficient at a given coupling strength results in the transition from spiral waves to traveling plane waves. In particular, we have demonstrated the emergence of spiral waves in the presence of the external stimulus. In contrast to previous findings, we discovered that the MTLN can result in the formation of spiral waves even in the absence of external stimuli. We believe that the obtained dynamical behavior of the MTLN may provide insight into how similar patterns might emerge in a variety of biological systems.

**Author Contributions:** Conceptualization, Z.N.T., A.K. and K.R.; methodology, B.S. and K.R.; software, B.S., Z.N.T. and K.R.; validation, A.K. and K.R.; formal analysis, B.S.; investigation, B.S. and Z.N.T.; resources, Z.N.T. and K.R.; data curation, B.S. and A.K.; writing—original draft preparation, B.S.; writing—review and editing, A.K. and K.R.; visualization, Z.N.T.; supervision, A.K. and K.R.; project administration, K.R.; funding acquisition, Z.N.T., A.K. and K.R. All authors have read and agreed to the published version of the manuscript.

**Funding:** This research received no external funding.

**Institutional Review Board Statement:** Not applicable.

**Informed Consent Statement:** Not applicable.

**Data Availability Statement:** Data generated during the current study will be made available at reasonable request.

**Acknowledgments:** We gratefully acknowledge that this work was funded by the Center for Nonlinear Systems, Chennai Institute of Technology (CIT), India, via funding number CIT/CNS/2022/RP-016.

**Conflicts of Interest:** The authors declare no conflict of interest.

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
