# Peer review of "Emerging Spiral Waves and Coexisting Attractors in Memductance-Based Tabu Learning Neurons"

_electronics, doi:10.3390/electronics11223685_

Round 1

Reviewer 1 Report

I have revised carefully this paper. Although the issue is interesting, this reviewer thinks that more details must be explained. Main for the cubic memductance which is not reflect the real behavior of a neuron. Further, the paper must be written following the template of the journal, here, the figures and equations are poorly positioned, making reading difficult. What are the contributions of section 3 and 4? Only analyzes and descriptions are mentioned here, but What about the comparison of the proposed neuron model with experimental biological measurements?

Author Response

We thank the Referee for providing a positive opinion of our manuscript.

We considered the cubic memductance in this study by referring to previous literature [1]-[5]. There is a large body of literature that considers the memristor function as g(φ) = α+3βφ2, and it is commonly used to realize the coupling between the membrane potential and magnetic flux.

We prepared the revised version of the manuscript in the Electronics journal template, as suggested by the Referee. The figures and equations are now properly aligned to make it easier for the readers to understand.

The dynamical properties of a single Tabu learning neuron are primarily discussed in Section 3 through numerical results. Sec. 4 provides the corresponding experimental validation. From the numerical and experimental results, we found the coexistence of chaotic and periodic behaviors. In addition, such characteristics are further validated through a bifurcation diagram, maximum Lyapunov exponents, and basin of attraction analysis.

The experimental biological measurements are part of the future scope of work, and we will perform them in the future. That is why we did not cover it in this study.

With these clarifications, we hope that we have clearly addressed all of the Referee’s comments and the Referee can now provide his/her recommendation of the revised manuscript for publication in Electronics.

[1] Wu, F., Zhang, G., & Ma, J. (2019). A neural memristor system with infinite or without equilibrium. The European Physical Journal Special Topics, 228(6), 1527-1534.

[2] Wang, Y., Ma, J., Xu, Y., Wu, F., & Zhou, P. (2017). The electrical activity of neurons subject to electromagnetic induction and Gaussian white noise. International Journal of Bifurcation and Chaos, 27(02), 1750030.

[3] Lv, M., Wang, C., Ren, G., Ma, J., & Song, X. (2016). Model of electrical activity in a neuron under magnetic flow effect. Nonlinear Dynamics, 85(3), 1479-1490.

[4] Chua L. Memristor: The missing circuit element. IEEE Trans Circuit Theor, 1971, 18: 507–519.

[5] Strukov D B, Snider G S, Stewart D R, et al. The missing memristor found. Nature, 2008, 453: 80–83.

Reviewer 2 Report

Please find the comment in the attachment.

Author Response

Point 1:  Correct the statement in the model description as "The coupling strength between the jth and kth neuron is given by σjk." Also, check the description of the state variables. • Explain the significance of the activation function in Tabu learning neurons.

Response 1:

We thank the Referee for this valuable suggestion. We corrected the statement, “The coupling strength between the jth and kth neuron is given by σjk.” in the revised manuscript as suggested by the Referee. Also, we provided the description for all the state variables. Further, the activation function plays a crucial role in exhibiting complex behaviors in neurons. In the revised manuscript, we also added the significance of the activation function in Tabu learning neurons.

Point 2:   Check Eqs. (5) and (7).

Response 2:

We appreciate the Referee’s suggestion. We have verified the Eqs. (5) and (7) and equations are given correctly.

Point 3:  Describe the novelty of this MTLN study in comparison to prior TLN findings.

Response 3:

The novelty of this study is provided in the conclusion section of the present version of the revised manuscript which is as follows:

  • The coexistence of chaotic and periodic behaviors is identified in Tabu learning neurons in the presence of magnetic flux.

  • We have provided experimental verification for the observed phenomena using a microcontroller-based TLN model.

  • Interestingly, the spiral wave pattern is observed even in the absence of external stimuli in contrast to the earlier findings.

Point 4: The section 4 discussion is insufficient. In particular, explain Fig. 5 in detail. 

Response 4:

We thank the Referee for this suggestion. We revised Section 4 in detail and provided a detailed explanation for Fig. 5 by adding the following statements in the revised manuscript.

“The microcontroller-based experiments offer more flexibility in order to make the portable source code into practice, customizing the system parameters, starting conditions, etc. It can also facilitate performing mathematical operations without the use of any special tools. Thus, we set up a system based on a microcontroller to experimentally mimic the observed coexisting dynamics of an MTLN. Figure 5(a) displays the corresponding schematic illustration. The microcontroller implementation is done in the first part using Arduino IDE and ESP32-WROOM-32D board. For the sake of simplicity, speed, and accuracy, the ESP32 microcontroller from ESPRESSIF. is used to discretize the system (3) and implement the tabu learning neuron model. Data acquisition can be done through the Arduino UNO. Finally out can be virtualized through the computer. The complete picture of the experimental setup for the microcontroller-based implementation of TLN is depicted in Fig. 5(b).

Figure 6 depicts the results of microcontroller implementation which clearly demonstrate that the experimental setup of Fig. 5 is capable of reproducing the coexisting behaviors observed in the tabu learning neuron model. By comparing Fig. 6 with Fig. 4(a), the experimental coexisting phase planes obtained from the microcontroller implementation match well with the numerical simulation equivalents. In this way, the results of the microcontroller implementation help us validate the dynamical characteristics of the proposed TLN model.”

Point 5: Explain why the periodic stimulus is applied in the lattice's center (see Fig. 8 description). What happens if it is applied to other nodes, whether or not the spiral wave appears? 

Response 5:

We appreciate the referee's insightful observation. The external periodic stimulus is typically used to activate each cell in an excitable medium, which then triggers the network's stable pulse or target waves. The periodic stimulus is used in this study at the center of the lattice array, which is the 75th node of j and k. When a periodic stimulus is applied to a node center, heterogeneity in energy levels can occur, causing nodes in the lattice to achieve lower and higher excitations, eventually resulting in a spiral wave pattern in the entire lattice. Periodic external stimuli to the edges or other specified nodes can also result in a spiral wave.

[1] Rajagopal, K., & Karthikeyan, A. (2022). Spiral waves and their characterization through spatioperiod and spatioenergy under distinct excitable media. Chaos, Solitons & Fractals, 158, 112105.

Point 6:  The citation format of references is inconsistent, which should be unified.

Response 6:

We thank the Referee for his suggestions on reference formatting. We provided all references in the unique manner suggested by the Referee in the amended text.

Point 7: Finally, the authors should double-check the manuscript for typographical and grammatical errors. 
 My overall comment is positive.

Response 7:

We appreciate the Referee's insightful suggestion. We double-checked the revised manuscript and carefully corrected all typographical and grammatical errors throughout the text.

With the above modifications and clarifications, we strongly believe that we have satisfactorily addressed all of the Referee's comments, and the Referee can now provide his/her recommendation of the revised manuscript for publication in Electronics.

Author Response

Point 1. In Eq. (11), there is a gap between the indexes j and k in some places but not in others. Similarly, the suffixes are misleading; k0 is frequently written as k0. Make it unique in the entire text.

Response 1:

We appreciate the Referee's useful observation. The gap between the indexes j and k in Eq. (11) and all other places are unified. Furthermore, in the revised manuscript, all suffixes are corrected and given uniquely.

Point 2. It is suggested that the authors modify the section titles and subtitles appropriately.

Response 2:

We thank the Referee for this suggestion. As suggested by the Referee, we have changed the section titles and subtitles suitably in the amended manuscript.

Point 3. Indexing is essential in Fig. 2. Nowhere in the text is the parameter m specified. However, a second figure for k0 vs m is displayed in Figure 2, and m is specified in the description for Figure 2. The authors are encouraged to double-check this.

Response 3:

We appreciate the Referee's helpful and informative remark.

  • The indexing is included in Fig.2.

  • "m" denotes the amplitude of the external driving current. In the revised manuscript, the expression for the external forcing current is modified as I_{ext} = m sin 2ft. We referred to the amplitude forcing curve as "m" rather than "A" in the entire text. In this manner, the discrepancy in the manuscript is corrected in the amended manuscript.

Point 4. Also, it would be interesting to provide some solid application for the findings reported in the paper. It is plenty of physical, chemical, biological, etc. oscillator networks reported in the literature: can the authors provide at least an example of a "real" system (with a thorough comparison, not just vague sentences) to which their model can be useful?

Response 4:

The application of this study is offered at the end of the conclusion section.

Point 5. The paper has some language issues which need to be checked and corrected in the revision.

Response 5:

The language issues in the manuscript have been rectified. We carefully corrected grammatical errors present in the manuscript.

Point 6. Highlight the novelty in the abstract and conclusion sections.

Response 6:

We highlighted the novelty of this study in the abstract and conclusion sections in the revised manuscript as suggested by the Referee.

With the above modifications and clarifications, we sincerely believe that we have taken care of all the comments of the Referee satisfactorily. We do hope that the Referee can now be able to recommend the publication of the revised manuscript in Electronics.

Reviewer 4 Report

In this article, authors have considered a modified Tabu learning neurons (MTLN) with magnetic flux. The obtained results are interesting from simulation and experimental points of view. A major revision is required to polish the manuscript results and findings better.

Author Response

Point 1. The whole manuscript is not formatted in a good manner; for instance, the mathematics portion is not well written, please rewrite all the equations and symbols in LaTeX or MathType, and all the figures should be placed in the middle of the pages, including the placement of figure captions, use (,) and (.) at the end of the equations, align the text from the left side to cover whole space, and parameters in mathematical form (in line 116-117).

Response 1:

We appreciate the Referee's recommendation on article formatting. To begin, we have provided the mathematical portions in MathType format, and the figures are properly aligned. We also added punctuation at the end of each equation. The manuscript is now uniquely structured in the Electronics journal format, as advised by the Referee.

Point 2. Use the “model” word after modified Tabu learning neurons, as it seems that the authors have constructed a model.

Response 2:

As indicated by the Referee, we used the term "model" after the modified Tabu learning neurons throughout the paper.

Point 3. Either use Sec. or Section no., according to the journal format.

Response 4:

As per the journal format, we used Sec. in the entire text uniquely.

Point 4: Please write the model description in detail including the meaning of parameters such as ?,?, ?0 ,?1 and ?2 .

Response 4:

As the Referee suggested, we added a detailed description of the model in the revised manuscript. The meanings of the parameters a, b, k0, k1, and k2 are also included.

Point 5:  The authors mentioned rest points; however, only positive rest point is shown somehow, which is not clear, explain.

Response 5:

We appreciate the Referee's insightful remark. The rest points equivalent system (4) has been confirmed and included in the updated paper.

Point 6: In Table 1, no value of ?0 is available where the system is stable. Why is that? Give the reason! If stability does not hold, then why prove stability?

Response 6:

We have presented the nature of stability around the origin for various magnitudes of k0 in the table. The suggested system shows an unstable focus for the given values of k0 which is confirmed by the eigenvalues.

Point 7: Describe the tool that is used to calculate the two-parameter bifurcation diagram in Figure 2 and provide a tool reference or citation.

Response 7:

We appreciate the Referee’s suggestion. Maximum Lyapunov exponents are used to obtain the two-parameter diagrams in Fig. 2 by varying the two distinct parameters simultaneously while keeping all other parameters constant. The color bar represents the maximum LE range that distinguishes the chaotic and periodic regions in the parametric spaces.

Point 8:  Describe more about the results calculated in the figures.

Response 8:

We appreciate the Referee's constructive suggestion. In the revised manuscript, we have provided a detailed description of the results calculated in each figure.

Point 9: Check the whole manuscript for grammatical errors, such as capitalization in Subsection 3.2, Fig.2, and Fig.3, etc.

Response 9:

The capitalization and grammatical errors in the entire manuscript have been rectified.

Point 10:  Arrange all the figure's scales properly, their length, width, etc.

Response 10:

We thank the Referee for this suggestion. The scales, length, and width of all the figures are modified according to the journal template.

Point 11:  Explain more in the conclusion section, besides the technical part.

Response 11:

The conclusion section has been improved in this revised manuscript. In particular, we included the novel finding of this study in the conclusion section.

Point 12. Compare the results with existing findings.

Response 12:

The effect of magnetic flux was not addressed in previous reports on Tabu leaning neurons. Furthermore, the coexistence of periodic and chaotic dynamics with electromagnetic flux has not been reported in 3D Tabu learning neurons. Furthermore, network dynamics are not well explored by the proposed neuron models.

In our manuscript, we addressed the aforementioned points by utilizing modified Tabu learning neurons with electromagnetic flux.

In the second paragraph of the introduction section, the existing findings of this study are discussed. Finally, the conclusion section highlights the obtained results in relation to the existing findings.

With the above modifications and clarifications, we sincerely believe that we have satisfactorily addressed all of the Referee's comments. The Referee can now recommend that the revised manuscript be published in Electronics.

Round 2

Reviewer 1 Report

Ready for acceptance from my point of view

Reviewer 3 Report

No further comments. 

Reviewer 4 Report

The authors have made the changes appropriately. The manuscript can be accepted.